# Prevalence of burnout among university students in low- and middle-income countries: A systematic review and meta-analysis

Mark Mohan Kaggwa[1], Jonathan Kajjimu[2]*, Jonathan Sserunkuma[2], Sarah Maria Najjuka[3], Letizia Maria Atim[1], Ronald Olum[3], Andrew Tagg[4,5], Felix Bongomin[6]

1 Department of Psychiatry, Faculty of Medicine, Mbarara University of Science and Technology, Mbarara, Uganda, 2 Faculty of Medicine, Mbarara University of Science and Technology, Mbarara, Uganda, 3 College of Health Science, Makerere University, Kampala, Uganda, 4 Emergency Department, Western Hospital-Footscray, Footscray, Victoria, Australia, 5 School of Medicine, University of Melbourne, Melbourne, Victoria, Australia, 6 Department of Medical Microbiology and Immunology, Faculty of Medicine, Gulu University, Gulu, Uganda

* jonathkebenz37@gmail.com

**Data Availability Statement:** All relevant data are within the paper and its Supporting information files.

## Abstract

### Background

Burnout is common among university students and may adversely affect academic performance. Little is known about the true burden of this preventable malady among university students in low-and-middle-income countries (LMICs).

### Objectives

This study aimed to systematically estimate the prevalence of burnout among university students in LMICs.

### Methods

We searched PubMed, Google Scholar, CINAHL, Web of Science, African Journals Online, and Embase from the inception of each database until February 2021. Original studies were included. No study design or language restrictions were applied. A random-effects meta-analysis was performed using STATA version 16.0. Heterogeneity and publication bias were assessed using Q-statistics and funnel plots, respectively.

### Results

Fifty-five unique articles, including a total of 27,940 (Female: 16,215, 58.0%) university students from 24 LMICs were included. The Maslach Burnout Inventory (MBI) was used in 43 studies (78.2%). The pooled prevalence of burnout was 12.1% (95% Confidence Interval (CI) 11.9–12.3; $I^2$ = 99.7%, Q = 21,464.1, $p$ = < 0.001). The pooled prevalence of emotional exhaustion (feelings of energy depletion), cynicism (negativism), and reduced professional efficacy were, 27.8% (95% CI 27.4–28.3; $I^2$ = 98.17%. $p$ = <0.001), 32.6 (95% CI: 32.0–

**Funding:** The authors received no specific funding for this work.

**Competing interests:** The authors have declared that no competing interests exist.

33.1; $I^2$: 99.5%; $p = < 0.001$), and 29.9% (95% CI: 28.8–30.9; $I^2$: 98.1%; $p = < 0.001$), respectively.

## Conclusion

Nearly one-third of university students in LMICs experience burnout. More studies are needed to understand the causes of burnout in this key population. There is a need to validate freely available tools for use in these countries.

## 1. Introduction

Burnout is a psychological syndrome that may arise as a response to chronic interpersonal stressors at work [1]. It is characterized by feelings of energy depletion and emotional exhaustion (EE). These feelings of exhaustion can be caused by educational demands, increased mental distance from one's studies, and feelings of cynicism related to one's studies/job. They may be coupled with reduced personal/professional efficacy (PE)—the feeling of incompetence as a student [2]. Burnout can happen to anyone involved in a psychologically engaging activity like higher education [3].

University education is an intrinsically demanding time in many students' lives[4–6]. There are several demands on a student's time including course work, relationships, examinations, part-time work, internship, pressure from parents and guardians, and practical/ward work for medical students [5, 6]. This puts many students at risk of burnout [5]. The literature has grouped burnout risk factors into 3 groups: *individual factors*, including sociodemographic variables; *education characteristics*, such as workload, time pressure, the course offered, part-time work, and *emotional demands* such as relationships; university characteristics. These include hierarchies, operating rules, resources, values, management model, culture, psychological support, and curricular factors [7–9]. Burnout is associated with poor academic performance, sleep disturbance, risk of severe mental illness or substance use disorder, an increased likelihood of cardiovascular disease, and neglect of physical and mental health [6, 10].

There have been disparities between recent studies [11–13] regarding the prevalence of burnout. In high-income countries (HICs), such as Saudi Arabia, the prevalence was noted 30.5% in a population of students doing healthcare-related courses [14]. In a similar cohort in Uganda, the prevalence was 54.5% [15].

Such discrepancies in prevalence rate were attributed to different exposures to socio-economic, political, health, and conflict-related stress factors [16]. Most systematic reviews on burnout have been performed among students doing health-related programs, predominantly in high-income countries (HICs) [17–20]. These overshadow the findings of students' wellness and burnout in low- and middle-income countries (LMICs). The few reviews done in LMICs are from middle-income countries (MICs) and have all been among students pursuing medical-related courses [21].

In this study, the research aimed to provide an insight into the burden of burnout syndrome among university students in LMICs by conducting a systematic review and meta-analysis to evaluate the prevalence of burnout and its different sub-components.

## 2. Methods

### 2.1 Study design

The researchers used the Meta-analysis of Observational Studies in Epidemiology (MOOSE) guidelines for systematic review and meta-analysis of observational studies [22], in addition to the Preferred Reporting Items for Systematic Reviews and Meta-Analyses (PRISMA) guidelines [23]. The study protocol was registered with PROSPERO (CRD42021232487).

### 2.2 Search strategy

With the help of a qualified medical librarian, relevant databases (PubMed, Google Scholar, CINAHL, Web of Science, African Journals Online, and Embase) were used for literature search, from the inception of each database until February 15[th], 2021. The search strings used were; burnout, burned out, emotional exhaustion, compassion fatigue, excess depersonalization, personal accomplishment, personal burnout, studies-related burnout, colleague related burnout, teacher-related burnout. Also, prevalence, burden, incidence, University Students, college students, Medical Students, and the list of all countries in Low Middle Income Countries (LMICs) [24] according to the World Bank Country and Lending Groups, 2021 (S1 File). Additional articles were obtained from a manual search of the references of the selected articles. The corresponding authors whose articles were not freely available were contacted by phone and via emails.

The articles included were published peer-reviewed articles of all languages, around the prevalence of burnout among university students in LMICs. Other languages were translated using Google translator. Cross-sectional, cohort, and case-control studies were included. Review articles, single case reports, and small case series were excluded.

### 2.3 Study selection process

All identified eligible articles were imported into Endnote 9 to ascertain duplicates. After removal of duplicates, two independent reviewers (LMA and JK) selected articles and abstracts. Any discrepancy in the included articles was settled by MMK. Articles were included for full article review by MMK and FB. The remaining articles were included for qualitative and quantitative synthesis checks. These articles were divided into pairs among (LMA and JS) and (JK and SMN). Any disagreement among the individuals was settled by the lead investigator (MMK), (Fig 1).

### 2.4 Data management and extraction

A pre-piloted Google form was used for data extraction. The following information were captured: The first authors, title, year of data collection, country, sample size, individuals with burnout and features of burnout, age, sex, year of study, the tool used, and study population.

### 2.5 Quality assessment

The risk of bias of the included articles was evaluated using the Joanna Briggs Institute (JBI) checklist [25]. JBI uses a 4-point Likert scale with answers being "no", "yes", "unclear", or "not applicable", for the following questions (1) appropriateness of the sample frame; (2) recruitment procedure; (3) adequacy of the sample size; (4) description of subjects and setting; (5) description of the identified sample; (6) validity of the methods used to screen for burnout; (7) reliability of the methods used to screen for burnout; (8) adequacy of statistical analyses; and (9) response rate. Articles were assigned one point per yes. Articles with a score less than 5

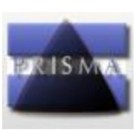

## PRISMA 2009 Flow Diagram

**Identification**

Records identified through database searching
(n = 2231)

Additional records identified through other sources
(n = 14)

**Screening**

Records after 295 duplicates removed
(n = 2214)

Records excluded
(n = 1956)
➢ Not reporting about burnout among students: 1633
➢ Conference papers: 26
➢ Review papers: 79
➢ Qualitative studies: 11
➢ Non peer reviewed articles: 130
➢ Citations: 7
➢ Others: 70

Records screened
(n = 2214)

**Eligibility**

Full-text articles assessed for eligibility
(n = 258)

Full-text articles excluded, with reasons
(n = 230)
➢ Studies done in high income countries: 155
➢ Not peer reviewed: 17
➢ Co-author did not provide full article: 5
➢ Study not done among university students: 19
➢ Studies not reporting prevalence of burnout among university students: 34
➢ Did not use standard instrument for measuring burnout: 1

**Included**

Studies included in qualitative synthesis
(n = 55)

Studies included in quantitative synthesis (meta-analysis)
(n = 55)

**Fig 1. The PRISMA flow diagram.** *From*: Moher D, Liberati A, Tetzlaff J, Altman DG, The PRISMA Group (2009). *Preferred Reporting Items for Systematic Reviews and Meta-Analyses: The PRISMA Statement.* PLoS Med 6(7): e1000097. doi:10.1371/journal.pmed1000097 **For more information, visit** www. prisma-statement.org.

were excluded. All the selected articles were included for further synthesis. The results are presented in S2 File.

## 2.6 Data analysis

Based on the number of students who had burnout, a random effect meta-analysis was performed using STATA version 16.0 (StataCorp, College Station, TX, USA). The Q statistics and $I^2$ accounted for the heterogeneity among the studies [26]. A random-effects model was used to determine the overall pooled estimates of effect sizes (prevalence), when heterogeneity was > 50, the researchers used fixed-effect models when the heterogeneity was lower. The results were presented on forest plots. The Funnel plots were generated to visually assess for publication bias. The study tools were summarized and a comparison between the different tools made concerning the relation to pooled burnout prevalence. P-values were two-tailed, and were considered statistically significant if the P-value was $\leq 0.05$.

## 3. Results

A total of 2245 studies were retrieved. Of these, 55 articles including a total of 28,206 (Male—11,121 and Female—16,398) university students and published between the year 2006 to 2020 met our inclusion criteria. Most of the studies were conducted in Brazil (n = 18), or China (n = 7). Many were from upper-middle-income countries (n = 45), with 10 from lower-middle-income countries and 3 from lower-income countries. The majority of studies were from South America (n = 30), Asia (n = 14), Africa (n = 10) and the least were from Europe (n = 3). A total of 3 studies were conducted among students undertaking non-medical related programs. 21 studies were conducted among postgraduate students. Participants were taken from the breadth of their training program. For details about the participants and study characteristics see Table 1.

## 3.1 Prevalence of burnout

55 studies reported the prevalence of burnout. 8,966 participants in LMICs had a burnout syndrome. The pooled prevalence, from the 24 countries, was 12.1% (95% CI 11.9–12.3; $I^2$ = 99.7%, Q = 21,464.1, $p$ = <0.001). There was marked heterogeneity between studies. The pooled prevalence in Upper Middle-Income countries (UMIC), Lower middle-income countries (LMIC), and Lower income countries was 9.8% (95% CI 9.6–10.0; $I^2$ = 99.7%, Q = 15444.6, $p$ = <0.001), 42.9% (95% CI 41.9–43.9; $I^2$ = 99.3%, Q = 1251.5, $p$ = <0.001), and 20.1% (95% CI 19.4–20.9; $I^2$ = 98.7%, Q = 156.8, $p$ = <0.001), respectively Fig 2. The funnel plot by countries' income status is presented in S3 File. Publication bias is seen especially among studies done in LMIC and LIC.

## 3.2 The prevalence of emotional exhaustion/feelings of energy depletion in LMICs

A total of 6,412 students displayed exhaustion, over 29 studies. The pooled prevalence was 27.8% (95% CI 27.4–28.3). There was significantly ($p$ = <0.001) high level of heterogeneity $I^2$ = 98.17%. and Q = 12,756.3. The pooled prevalence in UMIC, LMIC, and LIC was 30.1% (95% CI 29.4–30.7; $I^2$ = 99.8%, Q = 4,642.7, $p$ = <0.001), 8.6% (95% CI 7.8–9.4; $I^2$ = 99.8%, Q = 1,345.6, $p$ = <0.001), and 77.6% (95% CI 76.1–79.0; $I^2$ = 94.6%, Q = 36.8, $p$ = <0.001), respectively. Fig 3 shows the forest plot of the 30 studies and its funnel plot in S4 File.

**Table 1.** Characteristics of the studies.

| Author | Country | Year of data collection | Sample Size | Gender Male | Female | Age | Years of study | Study population | burnout n (%) | Emotional exhaustion n (%) | Cynicism n (%) | Reduced professional efficiency n (%) | The tool used to assess burnout |
|---|---|---|---|---|---|---|---|---|---|---|---|---|---|
| Franco et al., 2011 [27] | Brazil | 2004 | 16 | 1 | 15 | 25.8 | | P, M | 1 | 2 | 2 | 3 | MBI |
| Waldman et al., 2009 [28] | Argentina | 2007 | 106 | 70 | 36 | 29.1 | | P, M | 85 | 76 | 72 | 12 | MBI |
| Costa et al., 2012 [29] | Brazil | 2009 | 369 | 186 | 183 | | | U, M | 38 | 231 | 175 | 64 | MBI |
| Jovanović et al., 2016 [30] | Belarus | 2009 | 14 | 2 | 12 | 25.1(1.1) | | P, M | 5 | | | | MBI |
| Jovanović et al., 2016 [30] | Bosnia and Herzegovina | 2009 | 20 | 6 | 14 | 33.6(3.4) | | P, M | 3 | | | | MBI |
| Jovanović et al., 2016 [30] | South Africa | 2010 | 20 | 9 | 11 | 31.8(5.9) | | P, M | 12 | | | | MBI |
| Martins et al., 2011 [31] | Argentina | 2011 | 74 | 14 | 60 | | | P, M | 49 | | | | MBI |
| Nikodijević et al., 2012 [32] | Serbia | 2011 | 376 | 159 | 217 | - | - | U, N | 78 | 88 | 83 | 93 | MBI |
| Tavares et al., 2014 [33] | Brazil | 2011 | 48 | 4 | 44 | 26(2.9) | Year 2 (48) | P, M | 10 | 16 | 16 | 32 | MBI |
| Mason & Nel, 2012 [34] | South Africa | 2011 | 80 | 7 | 73 | 22.4 | year 1 (33), year 2 (24), year 3 (23) | U, M | 10 | | | | ProQOL R-IV |
| Mafla et al., 2015 [35] | Colombia | 2012 | 5647 | 1719 | 3928 | Range 18–24; Under 18 (665), 18–21 (2371), 22–24 (1793), >24 (818) | Year 1 (1348), Year 2 (1294), Year 3 (1178), Year 4 (1144), & Year 5 (683) | U, M | 394 | | | | MBI |
| Neves et al., 2016 [36] | Brazil | 2013 | 105 | | | 21.25(2.53) | | U, M | 11 | | | | MBI |
| Talih et al., 2016 [37] | Lebanon | 2013 | 118 | 62 | 56 | 18–25(26), 26–35 (92) | Year 1 (33), Year 2 (31), Year 3 (26), Year 4 (28) | P, M | 32 | | | | MBI |
| Bera et al., 2013 [38] | India | 2013 | 596 | 529 | 67 | | | U, M | 310 | | | | Not clear |
| Almeida et al., 2016 [39] | Brazil | 2013 | 376 | 151 | 216 | <21(119), 21–25 (209), 26> (39) | Year 1 (101), Year 2 (76), Year 3(100), Year 4(89) | U, M | 56 | | | | MBI |
| Parra-Osorio et al., 2015 [40] | Colombia | 2013 | 201 | 90 | 111 | 20.9 (2.9) | | U, M | 1 | | | | MBI |
| Pereira-Lima et al., 2017 [41] | Brazil | 2014 | 305 | 159 | 146 | 28(2.530) | Year 1& 2 (156), Year 3–5 (149) | P, M | 32 | 211 | 161 | 36 | OLBI |

(*Continued*)

Table 1. (Continued)

| Author | Country | Year of data collection | Sample Size | Gender Male | Female | Age | Years of study | Study population | burnout n (%) | Emotional exhaustion n (%) | Cynicism n (%) | Reduced professional efficiency n (%) | The tool used to assess burnout |
|---|---|---|---|---|---|---|---|---|---|---|---|---|---|
| Galdino et al., 2016 [42] | Brazil | 2014 | 129 | 13 | 116 | 32.3 | Masters (79), Ph.D. (50), | P, M | 15 | 90 | 35 | 32 | MBI |
| Tian et al., 2019 [43] | China | 2014 | 2008 | 593 | 1218 | | Masters' Year 1 (391), Master's Year 2 (554), Master's Year 3 (609), Ph.D. Year 1 (79), Ph.D. Year 2 (73), Ph.D Year 3 (68) | P, M | 1516 | | | | MBI |
| Pu et al., 2021 [44] | China | 2014 | 1814 | 596 | 1218 | | Master's Year 1 (391), Master's Year 2 (554), Master's Year 3 (609), Ph.D. Year 1 (79), Ph.D. Year 2 (73), & Ph.D. Year 3 (68) | P, M | 1482 | | | | MBI |
| Stein et al., 2016 [45] | South Africa | 2015 | 93 | | | | Year 4(93) | U, M | 29 | | | | CBI |
| Barbosa et al., 2018 [46] | Brazil | 2015 | 399 | 177 | 222 | 21(3.6) | - | U, M | 48 | | | | MBI |
| Fares et al., 2016 [16] | Lebanon | 2015 | 165 | 88 | 77 | 18–24 (161) >25 (4) | Year 1 (80), year 2 (85) | U, M | 124 | | | | MBI |
| Wickramasinghe et al., 2018 [47] | Sri Lank | 2015 | 796 | 356 | 440 | 18.4 (0.32) | - | U, N | 293 | | | | MBI |
| Fontana et al., 2020 [48] | Brazil | 2015 | 121 | 68 | 53 | 25 | Year 1 (33) & year 2 (35) | P, M | 67 | 38 | 52 | 46 | MBI |
| Liu et al., 2018 [49] | China | 2016 | 453 | 199 | 254 | 20.21(1.46) | Year1(129), Year2 (27) Year3(280), Year4 (16), Year5(1) | U, M | 42 | | | | MBI |
| Malik et al., 2016 [50] | Pakistan | 2016 | 133 | 98 | 35 | | Year1(46), Year2(45), Year3(22), Year4(19) | P, M | 77 | 67 | 66 | 27 | MBI |
| Wing et al., 2018 [6] | Malaysia | 2016 | 538 | 312 | 226 | 22.3 (1.3) | | U, M, N | 126 | | | | CBI |
| Mathias et al., 2017 [51] | South Africa | 2016 | 67 | 13 | 54 | 20–24(57), 25–29(5), 30-34(4), >35(1) | Year 3 (26), Year 4 (41) | U, M | 4 | | | | ProQOL R-IV |
| Atlam, 2018 [52] | Egypt | 2016 | 672 | 232 | 440 | <22 (411), >21 (261) | | U, M | 537 | | | | CBI |
| Serrano et al., 2016 [53] | Colombia | 2016 | 180 | 94 | 86 | 20(19–22) | | U, M | 18 | 66 | 67 | 46 | MBI |

(Continued)

Table 1. (Continued)

| Author | Country | Year of data collection | Sample Size | Gender Male | Gender Female | Age | Years of study | Study population | burnout n (%) | Emotional exhaustion n (%) | Cynicism n (%) | Reduced professional efficiency n (%) | The tool used to assess burnout |
|---|---|---|---|---|---|---|---|---|---|---|---|---|---|
| Lee et al., 2020 [54] | China | 2017 | 731 | 323 | 408 | 20.54 (2.07) | Year 1 (118), Year 2 (289), Year 3 (90), Year 4 (148), Year 5 (167), & Year 6 (37) | U, M | 204 | 360 | 393 | 520 | MBI |
| Calcides et al., 2019 [55] | Brazil | 2017 | 184 | 83 | 101 | 25.9(3.9) | | P, M | 66 | 98 | 96 | 35 | MBI |
| Haile et al., 2019 [56] | Ethiopia | 2017 | 144 | 123 | 98 | 30(3) | Year 1 (17), Year 2 (28), Year 3 (28), Year 4 (39), Year 5 (32) | U, M | 49 | 89 | 69 | 86 | MBI |
| Lopes et al., 2020 [57] | Brazil | 2017 | 284 | 28 | 256 | | | U, M | 17 | 103 | 107 | 80 | MBI |
| Tlili et al., 2021 [58] | Tunisia | 2017 | 368 | 49 | 319 | | Master's (40), others (328) | P, U, M | 252 | 130 | 75 | 128 | MBI |
| Vidhukumar & Hamza, 2020 [59] | India | 2017 | 375 | 142 | 233 | | Year 2 (60), Year 3 (73), Year 4 (34), Year 5 (44), & Interns (60) | U, M | 182 | | | | CBI |
| Vasconcelos et al., 2020 [60] | Brazil | 2017 | 100 | 9 | 91 | 18–27 (87), 27–57 (13) | Year 1 (36), Year 2 (16), Year 3 (24), & Year 4 (24) | U, M | 20 | 75 | 29 | 33 | MBI |
| Boni et al., 2018 [61] | Brazil | 2017 | 330 | 96 | 183 | 21.4(2.7) | Year1(118), Year2 (59), Year3 (51), Year4 (49) | U, M | 119 | 187 | 140 | 58 | MBI |
| Müller et al., 2020 [62] | Brazil | 2017 | 126 | 53 | 73 | 18–20 (32), 21–25 (81), 26–30 (11), & > 31 (2) | Year 1 (32), Year 2 (28), Year 3 (27), & Year (39) | U, M | 8 | 50 | 45 | 44 | MBI |
| Alhaffar et al., 2019 [63] | Syria | 2018 | 3350 | 1477 | 1873 | 21–25 (1139), 26–30 (1994), 31–35 (217) | Year 1 (1311), Year 2 (887), Year 3 (510), Year 4 (358), & Year 5 (284) | P, M | 646 | 2609 | 1829 | 2160 | MBI |
| Tavares et al., 2020 [64] | Brazil | 2018 | 419 | 148 | 271 | 22.1(4.3) | | U, M | 40 | 161 | 123 | 137 | MBI |
| Ji et al., 2020 [65] | China | 2018 | 380 | 79 | 301 | 26–29 (214), >29 (166) | Year 1 & 2 (160), Year 3 (73), Year 4 & 5 (147) | P, M | 233 | | | | MBI |
| Bolatov et al. 2021 [66] | Kazakhstan | 2018 | 771 | 193 | 578 | 20.7 | Year 1 (218), Year 2 (137), Year 3 (125), Year 4 (62), Year 5 (60), & Year 6 (169) | U, M | 216 | 451 | 429 | 98 | CBI |

(Continued)

Table 1. (Continued)

| Author | Country | Year of data collection | Sample Size | Gender Male | Gender Female | Age | Years of study | Study population | burnout n (%) | Emotional exhaustion n (%) | Cynicism n (%) | Reduced professional efficiency n (%) | The tool used to assess burnout |
|---|---|---|---|---|---|---|---|---|---|---|---|---|---|
| Daud et al., 2020 [67] | Malaysia | 2019 | 182 | 53 | 129 | | Year 1 (37), Year 2 (40), Year 3 (33), Year 4 (35), & Year 5 (37) | U, M | 67 | 100 | | 58 | CBI |
| Magri et al., 2019 [68] | Brazil | 2019 | 57 | 18 | 39 | 23(2,5) | | U, M | 5 | 19 | 19 | 19 | MBI |
| Pharasi et al., 2020 [8] | India | 2019 | 196 | 133 | 63 | 20.35 (1.50) | Year 1 (48), Year 2 (50), Year 3 (50), Year 4 (48) | U, M | 33 | 3 | 94 | 165 | MBI |
| Pokhrel et al., 2020 [69] | Nepal | 2019 | 651 | 496 | 156 | 25(4) | Year 1(66), Year2 (68), Year3 (71), Year4 (63), Year5 (63) | U, P, M | 318 | 266 | 210 | 105 | CBI |
| Aghajari et al., 2019 [70] | Iran | 2019 | 223 | | | | | U, M | 159 | | | | BABI |
| Geng et al., 2020 [71] | China | 2019 | 563 | 221 | 342 | <27 (215), >26 (348) | | P, M | 93 | 313 | 167 | 234 | MBI |
| Khosravi et al., 2021 [72] | Iran | 2019 | 400 | 156 | 244 | <24 (222), >23 (178) | Year 1 (68), Year 2 (58), Year 3 (80), Year 4 (77), & Others (117) | U, M | 102 | | | | BABI |
| Ogboghodo et al., 2020 [73] | Nigeria | 2019 | 448 | 279 | 169 | 33.9(4.0) | Year 1 (111), Year 2 (138), Year 3 (94), Year 4 (105) | P, M | 187 | | | | MBI |
| Kajjimu et al., 2021 [15] | Uganda | 2020 | 145 | 102 | 43 | 18–23 (91), 24–40 (54) | Year 1(22), Year 2 (28), Year 3 (38), Year 4 (26), Year 5 (31) | U, M | 79 | 135 | 141 | 90 | MBI |
| Zhang et al., 2021 [11] | China | 2020 | 684 | 290 | 234 | 20(17–24) | Year 2 (310), Year 3 (234), Year 4 (& above139) | U, M | 315 | 370 | 328 | 300 | LBS |
| Ogoma, 2020 [12] | Kenya | 2020 | 182 | 98 | 78 | 26.4(8.28) | Year 2 (95), Year 4 (50), Year 6 (37) | U, M | 32 | | | | MBI |
| Rodrigues et al., 2020 [13] | Brazil | 2020 | 350 | 160 | 169 | | Year 1 (66), Year 2 (62), Year 3 (57), Year 4 (63), Year 5 (50), Year 6 (55) | U, M | 13 | | | | MBI |
| Total | | | 279 | 10935 | 16215 | | | | 8966 | 6412 | 5031 | 4741 | |
| | | | 40 | | | | | | | | | | |

P = Postgraduate, U = Undergraduate, M = medical-related programs, N = Non-medical related programs, MBI = The Maslach Burnout Inventory, BABI = Breso Academic Burnout Inventory, CBI = Copenhagen Burnout Inventory, ProQOL R-IV = The fourth revision of the Professional Quality of Life Scale, OLBI = Oldenburg Burnout Inventory, and LBS = Learning Burnout Scale

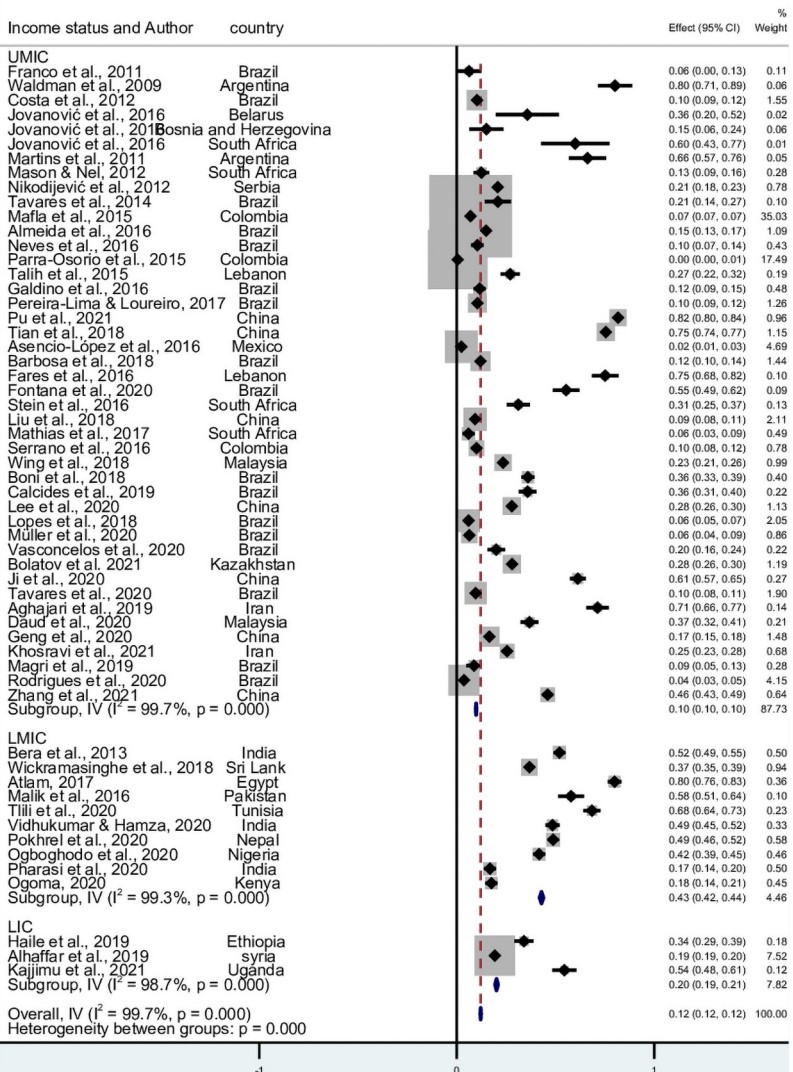

**Fig 2. Forest plot of the prevalence of burnout in LMICs.**

### 3.3 The prevalence of cynicism/negativism in LMICs

5,031 participants displayed high cynicism from 28 studies, conducted in 14 countries. The pooled prevalence was 32.6 (95% CI: 32.0–33.1; I²: 99.5%; Q = 5381.3, $p$ = <0.001). The forest plot in Fig 4 shows the distribution of the study prevalence's. S5 File shows a figure of the funnel plot of the studies.

### 3.4 The prevalence of reduced professional efficacy in LMICs

A total of 4,741 students had professional efficacy from 29 studies. The pooled prevalence was 29.9% (95% CI: 28.8–30.9; I² = 98.1%; Q = 1436.3, $p$ = <0.001). There was a high level of heterogeneity. Fig 5 shows the forest plot and accompanying funnel plot in S6 File.

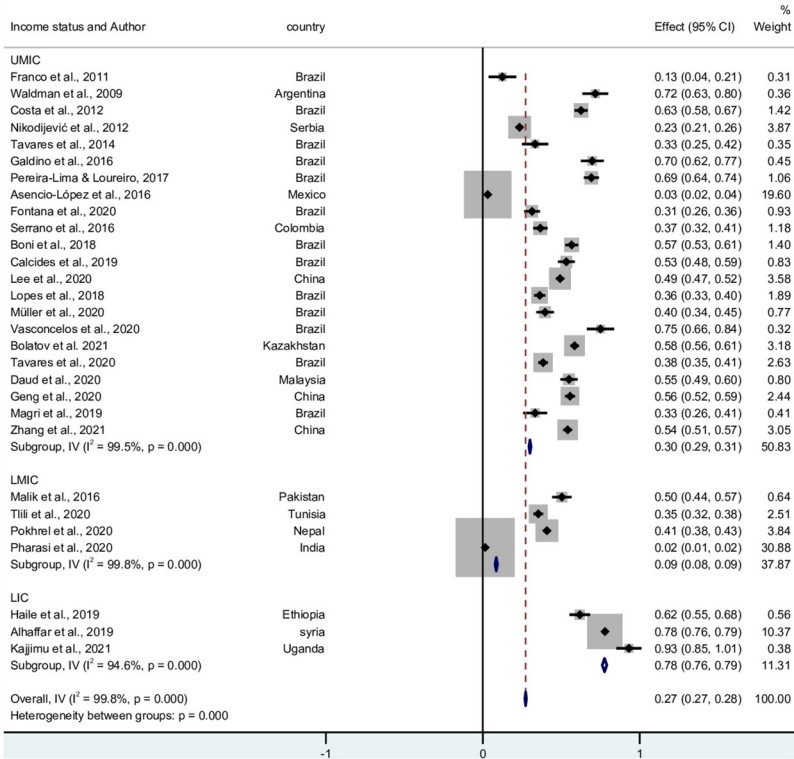

**Fig 3. Forest plot of the prevalence of emotional exhaustion in LMICs.**

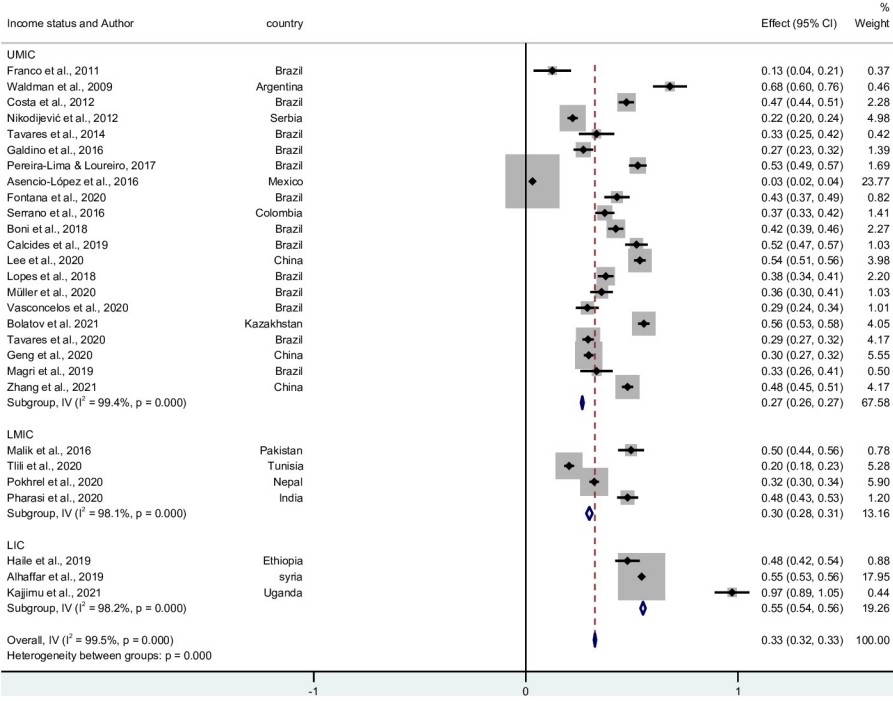

**Fig 4. Forest plot of the prevalence of cynicism in LMICs.**

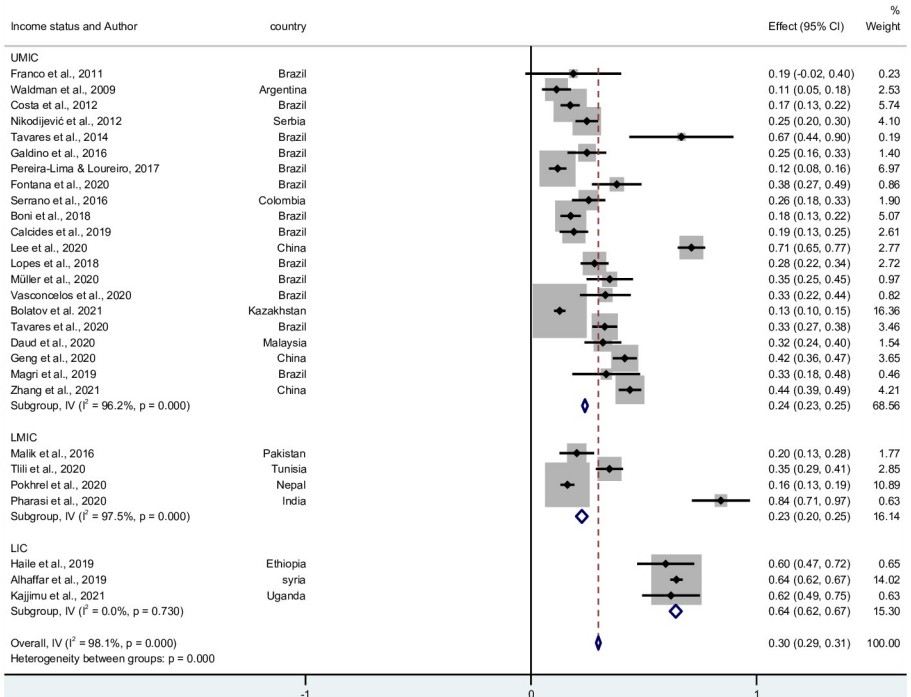

**Fig 5. Forest plot of reduced professional efficacy.**

## 3.5 The tools used to assess for burnout in LMICs

The most commonly used tool was the Maslach Burnout Inventory (MBI) (n = 42), followed by the Copenhagen Burnout Inventory (CBI) (n = 6). The Breso Academic Burnout Inventory and the fourth revision of the Professional Quality of Life (ProQOL R-IV) Scale were each used in 2 studies. The Oldenburg Burnout Inventory (OLBI), Learning Burnout Questionnaire, and the Russian version of the Copenhagen Burnout Inventory—Student Survey (R-CBI-S) were each used in one study. The methods are shown in Table 1

Disaggregated by the tool used, the pooled prevalence of burnout was 10.8% (95% CI: 10.6–11.0; $I^2$: 99.7%; Q = 16,051.5; $p$ = <0.001) using MBI, 41.6% (95% CI 40.3–42.9; $I^2$: 99.4%; Q = 823.7, $p$ = <0.001) using CBI; and 23.5 (95% CI 22.5–24.4: $I^2$ = 99.4; Q = 939.8, $p$ = <0.001) using other tools. Forest plot in S7 File.

## 3.6 Additional stratifications of burnout

**A. Region.** Burnout pooled prevalence was highest among African region, at 35.4% (95% CI: 34.1–36.7; Q = 1558.35; $I^2$ = 99.4%; $p$ = <0.001) from 10 studies, followed by the Asian region with a burnout pooled prevalence of 30.2% (95% CI: 29.7–30.6; Q = 7713.33; $I^2$ = 99.7%; $p$ = <0.001) from 22 studies, then followed by the European region with a burnout pooled prevalence of 20.7% (95% CI: 18.4–22.9; Q = 5.06; $I^2$ = 60.5%; $p$ = 0.080) from 3 studies. The South American region had the highest number of burnout studies but had the lowest burnout pooled prevalence of 5.9% (95% CI: 5.7–6.1; Q = 1990.92, $I^2$ = 98.9%; $p$ = <0.001) (S8 File).

**B. Field of study.** Only one study was done in both medical and non-medical students, the pooled prevalence of burnout was 2%, (95% CI: 21.3–25.5). Majority of the studies (n = 53)

were among medical students, and they had a burnout pooled prevalence of 12.1% (95% CI: 11.9–12.4; Q = 2396.12; $I^2$ = 99.7%, $p$ = <0.001). Two studies were done among non-medical students, but the pooled prevalence of burnout was similar to that among medical students i.e., 12.6% (95% CI: 12.4–12.8; Q = 97.75; $I^2$ = 99.0%; $p$ = <0.001) (S9 File).

**C. Level of study.** Most studies were done among undergraduate students and they had a burnout pooled prevalence of 9.1% (95% CI: 8.9–9.3; Q = 8634.67; $I^2$ = 99.6%; $p$ = < 0.001). Post graduate students had higher levels of burnout with a pooled prevalence of 29.2% (95% CI 28.7–29.8; Q = 6652.01; $I^2$ = 99.7%, $p$ = < 0.001) from 19 studies. Results indicating increasing burnout levels with increasing level of education. Two studies were conducted in both populations, with a burnout pooled prevalence of 54.5% (95% CI: 52.2–56.8; Q = 56.69; $I^2$ = 98.2%, $p$ = <0.001 (S10 File).

**D. Pre and during the COVID-19 pandemic.** A total of 53 stduies were conducted before the COVID-19 pandemic, with a burnout pooled prevalence of 12.2% (95% CI: 11.9–12.4; Q = 20349.64; $I^2$ = 99.7%, $p$ = < 0.001). During the COVID-19 pandemic, 4 studies were conducted whose pooled prevalence of burnout was 11.5% (95% CI: 10.1–11.9; Q = 1108.72; $I^2$ = 99.7%, $p$ = < 0.001).

## 4 Discussion

The objective of this systematic review and meta-analysis was to collate data surrounding the burden of burnout in university students in LMICs. The hope is that this can foster the implementation of evidence-based programs to combat burnout. To our knowledge, no other study has reviewed LMICs, and hence this study provides a great insight into levels of burnout, the tools used, and factors that could impact a student's academic performance. The results indicate a low level of burnout (12.0%), emotional exhaustion (27.8%), cynicism (32.6%), and reduced professional efficacy (29.9%). These findings suggest a lower rate than that found in high income country students as well as medical residents (26.8%–43.5%) or emergency medicine residents (55.6%– 77.9%), scoring for general burnout and 31.8%–46.0% for emotional exhaustion [17, 74, 75]. Our findings show evidence of marked publication bias. This could be a reason for the low levels of burnout reported in the region, as a result of few publications on the burnout aspects [76]. The findings were mainly from middle-income countries (MICs) (Brazil and China. The public are aware of the consequences of burnout [74, 77–79]. The low-income countries (LICs), which mainly reported a higher prevalence of burnout and its components [15, 30], need to preach more about burnout to increase the number of studies with country and cultural-based interventions to reduce burnout [80]. This unevenly distributed knowledge surrounding burnout may be related to the expensive study tools used in burnout screening. This precludes their use in LICs.

The MBI is most commonly used due to its good psychometric properties. It has been validated across many different cultures and countries [81]. Based on our synthesis, the prevalence of burnout among students, using the MBI tools, is statistically lower than that reported using other tools among other health workers [82]. The reliability of its findings appears comparable across different countries [81], but it currently remains the only tool that requires payment for use—a major hindrance in LMICs [83]. Countries should validate the freely available tools in order to have comparable results to those studies using the MBI tools. The shift to use of other instruments may increase knowledge and awareness of burnout in LMICs which could inform implementation of appropriate interventions.

This review shows that the level of burnout in LICs is higher compared to middle-income countries' economic levels among the LMICs. Perhaps, previously proven low-cost interventions among health workers should be used by students to improve coping skills and mitigate

education-related distress and burnout [84–86]. They include mindfulness practices, yoga exercises, and group discussions where individuals connect and share their experiences [84–87]. These are low-cost interventions, with online guides, that can be implemented in many LMICs. China and India have been using such methods to manage many conditions including depression, anxiety, other mental health challenges [88]. Introduction of these interventions among university students will require a context based approach to manage burnout. The role of universities in implementation of these interventions starts with the provision of education about the importance of the various interventions to all their students coupled with provision of coaches to guide students to perfect these self-administered interventions.

The African region had the highest pooled prevalence of burnout of university students at 35% (95% CI: 34–37) compared to any other region. Among health professional students, this may be due to unfavorable study conditions, high academic demands, and low training satisfaction. But whether this was due to having higher burnout in African university students or due to relatively fewer studies from Africa, remains a question to be answered by more research.

Our study found near identical pooled burnout prevalence between medical students (12.1%, 95% CI: 11.9–12.4) and non-medical students (12.6%, 95% CI: 12.4–12.8). A previous study demonstrated that overall burnout was more prevalent among medical students and residents than their age-matched colleagues not studying medicine [89]. This is most likely the underlying factor responsible for such a finding in our review because of the rigorous nature of medical training that health profession students have [15].

Our study found postgraduates to have a higher burnout pooled prevalence (29.2%, 95% CI: 28.7–29.8) compared to undergraduate students (9.1%, 95% CI: 8.9–9.3), further supporting the suggestion that the risk of students burning out increases with increase in the academic progression as previously found in a recent review by Dyrbye and Tait [7].

Our study found a lower pooled prevalence of burnout in students during the COVID-19 pandemic (11.5%, 95% CI: 10.1–11.9) compared to the burnout pooled prevalence (12.2%, 95% CI: 11.9–12.4) of students prior to the COVID-19 pandemic. Despite individual burnout studies conducted in the COVID-19 pandemic demonstrating high burnout prevalence [80]. This could most likely have occurred due to the few burnout studies done in the pandemic which we included in our synthesis, given the COVID-19 pandemic's adverse effect on wellbeing of students [90]. If more burnout descriptive studies could be conducted among university students during this pandemic, a clearer picture could be determined.

## 4.1 Strengths of the study

The literature search was done on several databases including articles in multiple languages. The studies all used standardized tools for measuring burnout. The study has a sub-analysis of burnout from different country income status, a previously not explored area by other meta-analyses.

## 4.2 Limitations

This review had several limitations. The authors included studies from both medical and non-medical students. They may not be comparable in the levels of burnout experienced. The study included postgraduate and undergraduate students who may have differing levels of stress. The different scoring methods used to determine burnout were not considered in this study. There was still marked heterogeneity of the results mainly among countries from different economic statuses.

### 4.3 Future direction

The researchers suggest the following to increase the understanding of burnout in LMICs. Longitudinal studies to identify the impact of education on student burnout. More studies are needed in non-medical students as they have been neglected in most of the studies. There is also a need to validate more tools in the different populations of LMICs to enable more reliable studies to be conducted in such an economic environment.

## 5 Conclusion

Burnout among university students in LMICs was low. The commonest screening tool used is the MBI and it showed lower burnout levels than other tools. Few studies have been conducted among university students especially in LICs and no observed variability in the use of other screening tools. This suggests the need for more studies to understand burnout and its associated factors in LMICs as well as validation of various freely available tools for use in these countries.

## Supporting information

**S1 File. Search strategy.**
(PDF)

**S2 File. Quality assessment of the included studies.**
(PDF)

**S3 File. The funnel plot for burnout prevalence.**
(PDF)

**S4 File. The funnel plot of the prevalence of emotional exhaustion in LMICs.**
(PDF)

**S5 File. Funnel plot of the prevalence of cynicism in LMICs.**
(PDF)

**S6 File. Funnel plot of the prevalence of professional efficacy in LMICs.**
(PDF)

**S7 File. Forest plot for the burnout measurement tools.**
(PDF)

**S8 File. Forest plot for the regional stratification of burnout pooled prevalence.**
(PDF)

**S9 File. Forest plot for the stratification of burnout pooled prevalence by field of study.**
(PDF)

**S10 File. Forest plot for the stratification of burnout pooled prevalence by level of study.**
(PDF)

**S11 File. Forest plot for the stratification of burnout pooled prevalence by the COVID-19 pandemic.**
(PDF)

**S1 Checklist. PRISMA checklist.**
(DOCX)

## Acknowledgments

Liberian Wilson Adriko who assisted with the literature search, The African Centre for Suicide Prevention and Research together with the CHINTA Research Bangladesh, which was formerly known as the Undergraduate Research Organization which provided training on the conduction of systematic review and meta-analysis.

## Author Contributions

**Conceptualization:** Mark Mohan Kaggwa, Jonathan Kajjimu.

**Data curation:** Mark Mohan Kaggwa, Jonathan Kajjimu, Jonathan Sserunkuma, Sarah Maria Najjuka, Letizia Maria Atim.

**Formal analysis:** Mark Mohan Kaggwa, Jonathan Kajjimu, Ronald Olum, Felix Bongomin.

**Funding acquisition:** Mark Mohan Kaggwa.

**Investigation:** Mark Mohan Kaggwa, Jonathan Kajjimu.

**Methodology:** Mark Mohan Kaggwa, Jonathan Kajjimu, Ronald Olum, Felix Bongomin.

**Project administration:** Mark Mohan Kaggwa, Jonathan Kajjimu, Letizia Maria Atim.

**Resources:** Mark Mohan Kaggwa, Jonathan Kajjimu.

**Software:** Mark Mohan Kaggwa, Jonathan Kajjimu.

**Supervision:** Mark Mohan Kaggwa, Jonathan Kajjimu, Andrew Tagg, Felix Bongomin.

**Validation:** Mark Mohan Kaggwa, Jonathan Kajjimu.

**Visualization:** Mark Mohan Kaggwa, Jonathan Kajjimu.

**Writing – original draft:** Mark Mohan Kaggwa, Jonathan Kajjimu, Letizia Maria Atim.

**Writing – review & editing:** Mark Mohan Kaggwa, Jonathan Kajjimu, Ronald Olum, Andrew Tagg, Felix Bongomin.

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
