## [Decision Letter · Decision Letter 0]

10 Jun 2021

PONE-D-21-12713

Prevalence of burnout among university students in low- and middle-income countries: A systematic review and meta-analysis.

PLOS ONE

Dear Dr. Kajjimu,

Thank you for submitting your manuscript to PLOS ONE. After careful consideration, we feel that it has merit but does not fully meet PLOS ONE’s publication criteria as it currently stands. Therefore, we invite you to submit a revised version of the manuscript that addresses the points raised during the review process.

This is an important topic, A few suggested edits are are follows:

1) Please correct the numbers in the PRISMA Flow Diagram. The numbers are not adding up. 2214 - 1886 = 328, not 258.

2) Please add a definition of LMIC, and mention which criterion for LMIC distribution did you use. If it was the World Bank, and please add it in the description. 

Please look at the reviewer comments.

We look forward to receiving your revised manuscript.

Kind regards,

Sabeena Jalal, MBBS, MSc, MSc, SM

Academic Editor

PLOS ONE

Journal Requirements:

2. Please amend the manuscript submission data (via Edit Submission) to include authorLetizia Maria Atim.

Reviewers' comments:

Reviewer's Responses to Questions

**Comments to the Author**

1. Is the manuscript technically sound, and do the data support the conclusions?

Reviewer #1: Yes

Reviewer #2: Yes

2. Has the statistical analysis been performed appropriately and rigorously? 

Reviewer #1: Yes

Reviewer #2: Yes

3. Have the authors made all data underlying the findings in their manuscript fully available?

Reviewer #1: Yes

Reviewer #2: Yes

4. Is the manuscript presented in an intelligible fashion and written in standard English?

Reviewer #1: No

Reviewer #2: Yes

5. Review Comments to the Author

Reviewer #1: The study presents the results of original research. Experiments, statistics, and other analyses are performed to a high technical standard and are described in sufficient detail. Also, the conclusions are presented in an appropriate fashion and are supported by the data. The research meets some of the applicable standards for the ethics of experimentation and research integrity and the article adheres to appropriate reporting guidelines and community standards for data availability.

Find the comments on the attached file.

Reviewer #2: The manuscript under consideration touches on an important topic of burnout among university students in LMIC. Below are a few comments to further strengthen the study.

1. Please add stratified results as suggested below:

a. Region

b. Field of study and

c. Level of study (grad vs undergrad) to address the limitation the authors mentioned.

Individual study results are not necessary, an overview table with aggregate results including basic information such as number of studies and prevalence would suffice. Please also add relevant methods and discussion.

2. Consider stratifying pre and during Covid19, if sufficient studies are available to separately look at this time period and add a brief discussion.

6. PLOS authors have the option to publish the peer review history of their article (what does this mean?). If published, this will include your full peer review and any attached files.

Reviewer #1: **Yes: **YAHAYA ABDULLAHI

Reviewer #2: No

---

## [Author Response · Author response to Decision Letter 0]

29 Jun 2021

Thanks you so much for all the helpful comments you gave to our paper. 

Editor’s comments

1. Please correct the numbers in the PRISMA Flow Diagram. The numbers are not adding up. 2214 - 1886 = 328, not 258.

Our response: 

Thanks for this keen observation to identify this issue. We had accidentally left out of the PRISMA flow diagram other papers (n=70) which we hadn’t categorized. 

2. 2) Please add a definition of LMIC, and mention which criterion for LMIC distribution did you use. If it was the World Bank, and please add it in the description

Our response: 

In the Methods section under the search strategry, we have added a LMIC definition of “Low and Middle Income Countries” according to the World Bank Country and Lending Groups, 2021. (reference 20)

 Our response: 

We have used these templates to make edits to our manuscript to enable it meet PLOS ONE’s style requirements.

2. Please amend the manuscript submission data (via Edit Submission) to include authorLetizia Maria Atim.

Our response: 

We have added Letizia Maria Atim as a coauthor.

Reviewer 1: 

4. Is the manuscript presented in an intelligible fashion and written in standard English?

Reviewer #1: No 

Our response: 

Our revised manuscript has been copy edited by a coauthor of the paper who is a native English speaker, called Dr. Andrew Tagg.

Comments in attached file.

Our response: 

We have responded to all comments in the attached file. 

Reviewer 2: 

1. Please add stratified results as suggested below:

a. Region

b. Field of study and

c. Level of study (grad vs undergrad) to address the limitation the authors mentioned.

Individual study results are not necessary, an overview table with aggregate results including basic information such as number of studies and prevalence would suffice. Please also add relevant methods and discussion.

Our response: 

We have included additional stratifications of our study data basing on the region, field of study, and level of study. Forest plots of these stratifications have also been supplied as supplementary files 9,10, and 11 respectively. 

2. Consider stratifying pre and during Covid19, if sufficient studies are available to separately look at this time period and add a brief discussion.

Our response: 

We have added a stratification of our data based on findings pre and during COVID-19. A supplementary file12 of a forest plot of this stratification has also been provided.

---

## [Decision Letter · Decision Letter 1]

6 Aug 2021

Prevalence of burnout among university students in low- and middle-income countries: A systematic review and meta-analysis.

PONE-D-21-12713R1

Dear Dr. Kajjimu,

We’re pleased to inform you that your manuscript has been judged scientifically suitable for publication and will be formally accepted for publication once it meets all outstanding technical requirements.

Kind regards,

Sabeena Jalal, MBBS, MSc, MSc, SM

Academic Editor

PLOS ONE

Additional Editor Comments (optional):

Reviewers' comments:

Reviewer's Responses to Questions

**Comments to the Author**

1. If the authors have adequately addressed your comments raised in a previous round of review and you feel that this manuscript is now acceptable for publication, you may indicate that here to bypass the “Comments to the Author” section, enter your conflict of interest statement in the “Confidential to Editor” section, and submit your "Accept" recommendation.

Reviewer #2: All comments have been addressed

2. Is the manuscript technically sound, and do the data support the conclusions?

Reviewer #2: Yes

3. Has the statistical analysis been performed appropriately and rigorously? 

Reviewer #2: Yes

4. Have the authors made all data underlying the findings in their manuscript fully available?

Reviewer #2: Yes

5. Is the manuscript presented in an intelligible fashion and written in standard English?

Reviewer #2: Yes

6. Review Comments to the Author

Reviewer #2: The authors sufficiently revised the manuscript to reflect all comments. Presentation of the stratified results strengthens the study.

7. PLOS authors have the option to publish the peer review history of their article (what does this mean?). If published, this will include your full peer review and any attached files.

Reviewer #2: No

---

## [Editor Report · Acceptance letter]

20 Aug 2021

PONE-D-21-12713R1 

Prevalence of burnout among university students in low- and middle-income countries: A systematic review and meta-analysis 

Dear Dr. Kajjimu:

I'm pleased to inform you that your manuscript has been deemed suitable for publication in PLOS ONE. Congratulations! Your manuscript is now with our production department. 

Kind regards, 

on behalf of

Dr. Sabeena Jalal 

Academic Editor

PLOS ONE